# Association between Clinical Periodontal Features and Glycated Hemoglobin in Patients with Diabetes and Controlled Periodontitis: A Cross-Sectional Study

**DOI:** 10.3390/healthcare11071035

**Published:** 2023-04-04

**Authors:** Masayuki Tooi, Yuhei Matsuda, Hui Zhong, Shinichi Arakawa

**Affiliations:** Department of Lifetime Oral Health Care Science, Graduate School of Medical and Dental Sciences, Tokyo Medical and Dental University, 1-5-45 Yushima, Bunkyo-ku, Tokyo 113-8510, Japan

**Keywords:** HbA1c, type 2 diabetes mellitus, periodontitis, periodontal inflamed surface area, probing pocket depth, supportive periodontal therapy

## Abstract

Diabetes and periodontitis are the most prevalent chronic diseases, and they influence each other’s progression. Only a few studies have shown the association between diabetes and mild periodontitis. We aimed to investigate the relationship between well-controlled periodontitis and glycated hemoglobin (HbA1c) in patients with diabetes. This retrospective study investigated 150 Japanese patients with type 2 diabetes treated with supportive periodontal therapy (SPT). Medical histories of diabetes and periodontal therapy were collected, and a multiple linear regression analysis was performed to determine their association. The patients included 67 (44.7%) males and 83 (55.3%) females, with a mean age of 68.1 (standard deviation = 10.5) years. Forty-four (29.3%) patients were treated for diabetes, and the mean HbA1c was 6.7% (0.7). Oral status was 23.3 (5.1) for the number of teeth, 2.5 mm (0.4) for mean probing pocket depth (PPD), and 163.9 mm^2^ (181.3) for the periodontal inflamed surface area (PISA). The multiple regression analysis showed a significant association between mean PPD (β = 0.38, *p* = 0.03) and HbA1c in patients with mild diabetes but not in severe cases. These results suggest that the diagnostic indices for periodontitis used to assess the association between periodontitis and HbA1c would be determined based on the severity of periodontitis and type 2 diabetes.

## 1. Introduction

Diabetes mellitus (DM) is a metabolic disease characterized by high blood glucose levels due to reduced insulin secretion or action [1]. It is often accompanied by complications, such as retinopathy, nephropathy, neuropathy, and cardiovascular diseases [2]. If not treated, patients may develop serious complications without experiencing any symptoms. DM also leads to the progression of atherosclerosis and an increased incidence of heart diseases, stroke, and deaths [3]. DM is one of the most prevalent chronic diseases worldwide. It was reported that the proportion of adults with diabetes globally reached 10.5% (536.6 million people) in 2021 and may increase to 12.2% (783.2 million) by 2045 [4]. Glycated hemoglobin (HbA1c) is an index of the percentage of hemoglobin-glucose complexes in the blood and indicates the risk of developing DM-related complications. It is an important biomarker used as a reference for clinical diagnostic criteria and a quantitative result of DM treatment [5]. Japanese clinical guidelines for diabetes consider HbA1c < 6.3% as the cut-off value [6].

Periodontitis is the most widespread infectious and chronic immuno-inflammatory disorder worldwide [7]. Epidemiological studies have demonstrated that severe periodontitis affects at least 43 systemic disorders, including vascular diseases, adverse pregnancy outcomes (preterm births and low birth weight), aspiration pneumonia, rheumatoid arthritis, and nonalcoholic steatohepatitis [8,9]. Notably, the relationship between DM and periodontal disease has been well studied. Patients with diabetes have a significantly higher incidence of periodontitis than those without diabetes [10]; additionally, patients with severe periodontitis experience a significantly higher incidence of type 2 DM and glucose intolerance [11,12]. Consequently, there is a mutual relationship between periodontitis and type 2 DM.

HbA1c is a useful and important index for type 2 DM. In contrast, periodontitis has many diagnostic indices, such as probing pocket depth (PPD), bleeding on probing (BOP), tooth mobility, tooth loss, clinical attachment loss (CAL), and radiographic evidence of marginal alveolar bone loss. Periodontitis is diagnosed by combining these data [13,14].

Physicians and dentists should cooperate and share information to treat diabetes and periodontitis successfully. One of the quantitative parameters for evaluating periodontitis is the periodontal inflamed surface area (PISA). PISA is calculated from the periodontal epithelial surface area (PESA) and BOP measurements, which are indicators of the tissue inflammatory response in the periodontal pocket epithelium [15]. Some reports have indicated that HbA1c and PISA are positively correlated. In patients with type 2 DM who had not received periodontal treatment for 6 months and those with severe periodontitis with poor glycemic control, PISA was positively correlated with HbA1c [16,17,18]. However, the association between HbA1c and PISA has been controversial. Conversely, Susanto et al. reported no association between HbA1c and PISA in patients with mild periodontitis receiving treatment for type 2 DM [19].

Supportive periodontal therapy (SPT) is a procedure for ensuring the long-term stability of the periodontal tissue after periodontal therapy. Dentists and dental hygienists perform dental hygiene instruction, periodontal pocket examination to check for progression/recurrence of periodontitis, and the removal of intraoral bacteria to reduce the possibility of reinfection and extended disease progression [20]. The continuation of this procedure will allow the maintenance of controlled periodontitis.

To allow long-term stable maintenance of controlled periodontitis, it may be important to monitor the status of DM during SPT given the mutual relationship between periodontitis and type 2 DM. However, the relationship between the clinical features of controlled periodontitis and HbA1c during SPT remains unclear. Therefore, it important to identify a periodontitis-related marker that can be used to assess the association between HbA1c levels and periodontitis during SPT. We hypothesized that PISA would be positively associated with HbA1c, even in patients with controlled periodontitis. This study aimed to examine the relationship of HbA1c levels with PISA and PPD in patients with DM undergoing SPT. Our findings could facilitate cooperation between dentists and physicians in the management of patients with diabetes and periodontitis.

## 2. Materials and Methods

### 2.1. Study Participants

This was a retrospective study included from patients treated at the Department of Oral Health Care, Dental Hospital of Tokyo Medical and Dental University (TMDU) (Tokyo, Japan), between April 2016 and May 2022. In this department, patients were treated with more than one SPT. Finally, 150 candidates diagnosed with type 2 DM at the Medical Hospital of TMDU (Tokyo, Japan) or other hospitals were finally included. We excluded hospitalized patients; patients undergoing cancer treatment; patients who underwent only one oral care session preoperatively; patients who underwent antibiotic therapy or oral surgery within the last 6 months; pregnant or lactating women; patients with drug-induced gingival overgrowth, which could lead to underestimation of the PISA [15]; and patients with severe immunodeficiency. This study was conducted per the Helsinki Declaration of 1975, revised in 2013, and approved by the local ethics committee of TMDU (D2022-021). As this was a retrospective study, informed consent was not required. The research protocol was published on our laboratory website and a bulletin board at the Department of Oral Health Care, Dental Hospital of TMDU, per the guaranteed opt-out option.

### 2.2. Clinical Data Collection

The following variables were collected as background data: age (years) and sex (male: female). Electronic medical records at the Medical Hospital of TMDU or blood test data from other hospitals were used to collect HbA1c data and the medical history of diabetes therapy. We analyzed HbA1c data recorded during the most recent SPT. Concerning diabetes therapy at other hospitals, medical histories were obtained from past medical interview records. The presence or absence of antidiabetic agent use was confirmed using the medication records. The patients were divided into two groups, with 6.3% as the cut-off value of HbA1c, according to Japanese clinical guidelines [6]. Other necessary information was obtained from past interview records. Full-mouth PPD was measured using PCP-UNC 15 or PCP 10 probes (Hu-Friedy, Chicago, IL, USA). An experienced periodontist or dental hygienist measured PPD and counted BOP at six sites on all residual teeth in the periodontal pocket examination. Simultaneously, the existing teeth were counted. In cases where the distal aspect of the second molar had PPD ≥ 4 mm and the next third molar was impacted or partially impacted, the distal PPD score at the second molar was eliminated [21]. The BOP score was calculated based on the bleeding count in pockets during the periodontal pocket examination and presented as a percentage of all examined sites. To quantitatively measure the spread of inflammation in the entire periodontal tissue, PESA and PISA were calculated following the report by Nesse et al. [15]. PESA for each tooth was calculated by multiplying the PPD by a constant determined for each pocket [22]. Moreover, PISA for each tooth was calculated by multiplying PESA with the proportion of BOP. The total PESA and PISA scores were calculated by summing the scores for each tooth [15]. The PESA and PISA scores were determined using the PPD and BOP at six sites on all residual teeth. Finally, we calculated the average PPD, BOP rate, PESA, and PISA using the JSP-Chart Ver4 software (FOD Inc., Takamatsu, Japan), with the PPD and BOP being entered to calculate PESA and PISA.

### 2.3. Statistical Analyses

To confirm the normality of data distribution, a Shapiro–Wilk test was performed. The number of participants (%) and mean (standard deviation (SD)) were calculated for each participant’s data as descriptive statistics. The analysis was performed following two stages.

Multivariate linear regression was used to control for the possible confounding effects of variables related to HbA1c, and the partial regression coefficient for each HbA1c outcome was estimated after adjusting for all other variables included in the model. Adjustments were made for the following variables: sex, age, number of teeth, and mean PPD in model 1; and sex, age, number of teeth, and PISA in model 2. Since this study was a retrospective study, the sample size was not calculated.

All statistical analyses were performed using SPSS version 26 (IBM SPSS Japan Inc., Tokyo, Japan). Two-tailed *p*-values were calculated for all analyses. The alpha level of significance was set at 0.05.

## 3. Results

### 3.1. Participants’ Characteristics

We enrolled 150 patients who met the eligibility criteria in this study. The patients included 67 (44.7%) males and 83 (55.3%) females, with a mean age (SD) of 68.1 (10.5) years. Forty-four (29.3%) patients were treated for diabetes; the mean HbA1c level was 6.7% (0.7). Oral status was 23.3 (5.1) for the number of teeth, 2.5 (0.4) for mean PPD, and 163.9 (181.3) for PISA. The demographic characteristics of the participants are summarized in Table 1.

### 3.2. Multivariate Linear Regression Analysis of the Association between PPD or PISA and HbA1c in Patients with Mild Diabetes Mellitus

Multiple regression analysis showed a significant association between the mean PPD and mild diabetes (β = 0.38, *p* = 0.03) in model 1. Model 2 did not show any significant associations. Details of the simple and multiple regression analyses are presented in Table 2.

### 3.3. Multivariate Linear Regression Analysis of the Association between PPD or PISA and HbA1c in Patients with Severe Diabetes Mellitus

Multiple regression analysis showed no significant association between PISA and HbA1c in models 1 and 2 among patients with severe diabetes mellitus. Details of the simple and multiple regression analyses are presented in Table 3.

## 4. Discussion

This study analyzed the relationship between HbA1c and diagnostic indices for periodontitis: BOP, PPD, and PISA. We found that HbA1c and PPD had a significant positive association among patients with mild diabetes undergoing SPT. Previous reports described that exacerbation of periodontitis was observed in patients with diabetes who, consequently, had deeper periodontal pockets [23,24]. In epidemiological studies among Japanese individuals, Morita et al. reported that patients with type 2 DM were 1.17 times more likely to experience periodontal tissue destruction than healthy individuals. They concluded that diabetes affected the deterioration of the periodontal pockets [25]. In addition, the Hisayama cross-sectional study revealed that periodontal pocket depth was significantly associated with impaired glucose tolerance and diabetes. The average HbA1c level of patients with DM was 6.4%, close to that in our study (6.7%) [11]. Moreover, this retrospective cohort study clarified that participants who had developed glucose intolerance or diabetes during the past 10 years showed a significant increase in periodontal pocket depth on subsequent examination [11]. These results suggest that considering the association between PPD increase and the development of diabetes is important.

Moreover, we could not find an association between PISA and HbA1c regardless of the severity of diabetes. PISA is one of the clinical parameters used to evaluate periodontal conditions, particularly inflamed surface areas in periodontal tissues [16]. The mean value of PISA in our study was much smaller than that in previous reports, which reported an association between PISA and HbA1c [17,18]. As our study participants were in the SPT phase and had little gingival inflammation, the PISA values were low and indicated mild periodontitis [26]. This result implied that there was no association between PISA and HbA1c, partially because fewer patients had severe diabetes in our study than in previous reports. The average HbA1c level was approximately 7–9% in some studies on the association between PISA and HbA1c [16,17,18], higher than the average of 6.7% obtained in our study. Therefore, by expanding the research to include patients with higher HbA1c levels, we will better understand the association between PISA and PPD and HbA1c and set up cut-off values for diabetes treatment.

Furthermore, the PESA is defined by a six-order function with the PPD as a parameter, increasing monotonically within the PPD measurement range. PISA is proportional to the number of BOP as it is calculated by multiplying PESA with the percentage of BOP [15,22]. Moreover, the PISA formula includes two parameters: PPD and BOP; however, based on the PISA calculation, BOP could greatly influence the score results. In our study, PPD was associated with HbA1c; however, there was no association between BOP and HbA1c levels (data not shown). As the study participants were limited to patients with relatively well-controlled periodontal inflammation, we considered the BOP levels too small to increase PISA values; the relationship between PISA and HbA1c was not observed.

Our study results provide an index for detecting the onset and progression of type 2 DM in patients with mild diabetes in the SPT phase. Although we observed no association between HbA1c and clinical parameters of periodontitis in severe diabetes, PISA may predict HbA1c levels in patients with severe diabetes who have extensive inflammation in the periodontal tissue [17,18]. As described above, to measure the association between periodontitis and HbA1c, different diagnostic indices should be used for periodontitis while considering the degree of periodontitis and DM. By constructing this step-by-step system, information about DM and periodontitis could be easily shared between physicians and dentists, facilitating successful treatments.

In addition to basic periodontal therapy for periodontitis in patients with diabetes, other useful non-surgical adjunctive therapies include melatonin, laser, and doxycycline, which have relatively few side effects and can be used safely in patients with systemic diseases, including DM [27,28,29]. Even in the SPT phase, it is important to treat worsening periodontitis and DM, and our study may contribute to the examination of the effectiveness of such treatment options.

This study has several limitations. First, the causality of the association is unknown because it was a cross-sectional study. Second, no adjustments were made based on DM therapy because there were no data regarding the details of the medications being administered for DM. Third, the number of patients with a smoking history was unclear due to missing data. Fourth, there is a potential for healthy volunteer bias. Fifth, the risk of developing DM is reported to be higher in males than in females [6]; however, most of our participants were females. Sixth, the weight records of our patients were unclear because of missing data. A high BMI and obesity increase the risk of insulin resistance [6,19]. These parameters should be considered in future studies as risk factors for DM. Therefore, larger longitudinal studies in a population of patients with mild diabetes and controlled periodontal disease are warranted for future research.

## 5. Conclusions

HbA1c was associated with mean PPD higher among patients with mild-type 2 DM treated with SPT. The results of this study suggest that different diagnostic indices for periodontitis should be used when assessing the association between periodontitis and HbA1c according to the degree of periodontitis and type 2 DM. On the other hand, we did not observe an association of HbA1c with PISA or other clinical parameters of periodontitis in severe diabetes. More detailed studies are warranted to define the adaptation range of PISA.

## Figures and Tables

**Table 1 healthcare-11-01035-t001:** Demographic data (N = 150).

Item	Category	N (%) or Mean (Standard Deviation)
Age (years)		68.1 (10.5)
Sex	Male	67 (44.7)
	Female	83 (55.3)
DM treatment (yes)		44 (29.3)
HbA1c		6.7 (0.7)
Number of teeth		23.3 (5.1)
PPD		2.5 (0.4)
PISA		163.9 (181.3)

DM, diabetes mellitus; PPD, probing pocket depth; and PISA, periodontal inflamed surface area.

**Table 2 healthcare-11-01035-t002:** Multivariate analysis of the association between PPD or PISA and HbA1c in patients with mild diabetes mellitus. (N = 37).

Model 1												
Variables	Univariate						Multivariate					
			95% CI						95% CI			
	β	B	lower	upper	*p*		β	B	lower	upper	* **p** *	Adjusted R^2^
Age	0.23	0.01	−0.003	0.02	0.18		0.35	0.01	0.0002	0.02	0.04*	0.14
Sex	0.05	0.03	−0.19	0.26	0.78		0.16	0.10	−0.12	0.32	0.94	
Number of teeth	0.12	0.01	−0.02	0.03	0.47		0.21	0.02	−0.01	0.04	0.24	
Mean PPD	0.31	0.24	−0.01	0.49	0.06		0.38	0.29	0.04	0.55	0.03*	
**Model 2**												
**Variables**	**Univariate**						**Multivariate**					
			**95% CI**						**95% CI**			
	**β**	**B**	**lower**	**upper**	*p*		**β**	**B**	**lower**	**upper**	* **p** *	**Adjusted R^2^**
Age	0.23	0.01	−0.003	0.02	0.18		0.29	0.008	−0.002	0.02	0.12	0.002
Sex	0.05	0.03	−0.19	0.26	0.78		0.08	0.05	−0.18	0.29	0.64	
Number of teeth	0.12	0.01	−0.02	0.03	0.47		0.23	0.02	−0.01	0.05	0.24	
PISA	0.15	0.0002	−0.0002	0.0007	0.39		0.11	0.0002	0.0003	0.001	0.54	

CI, confidence interval; PPD, probing pocket depth; and PISA, periodontal inflamed surface area.

**Table 3 healthcare-11-01035-t003:** Multivariate analysis of the association between PPD or PISA and HbA1c in patients with severe diabetes mellitus. (N = 113).

Model 1												
Variables	Univariate						Multivariate					
			95% CI						95% CI			
	β	B	lower	upper	*p*		β	B	lower	upper	*p*	Adjusted R^2^
Age	−0.15	−0.01	−0.02	0.002	0.12		−0.17	−0.01	−0.02	0.002	0.09	0.008
Sex	−0.09	−0.10	−0.33	0.12	0.36		−0.07	−0.08	−0.31	0.14	0.46	
Number of teeth	−0.07	−0.008	−0.03	0.01	0.43		−0.10	−0.01	−0.03	0.01	0.34	
Mean PPD	0.08	0.12	−0.16	0.40	0.40		0.06	0.09	−0.19	0.38	0.53	
**Model 2**												
**Variables**	**Univariate**						**Multivariate**					
			**95% CI**						**95% CI**			
	**β**	**B**	**lower**	**upper**	* **p** *		**β**	**B**	**lower**	**upper**	* **p** *	**Adjusted R^2^**
Age	−0.15	−0.01	−0.02	0.002	0.12		−0.19	−0.01	−0.02	0.001	0.06	0.01
Sex	−0.09	−0.10	−0.33	0.12	0.36		−0.06	−0.08	−0.30	0.150	0.51	
Number of teeth	−0.07	−0.008	−0.03	0.01	0.43		−0.11	−0.01	−0.03	0.009	0.26	
PISA	−0.06	−0.0002	−0.001	0.0004	0.53		−0.08	−0.0002	−0.001	0.0003	0.39	

CI, confidence interval; PPD, probing pocket depth; and PISA, periodontal inflamed surface area.

## Data Availability

All of the data were corrected in this research. Derived data supporting the findings of this study are available from the author (M.T.) on request.

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
