# Peer review of "Association between Clinical Periodontal Features and Glycated Hemoglobin in Patients with Diabetes and Controlled Periodontitis: A Cross-Sectional Study"

_healthcare, 2023, doi:10.3390/healthcare11071035_

Round 1

Reviewer 1 Report

First of all, I would like to greet the authors and congratulate them on the theme and work done. The study appears correctly performed and written without logical or factual errors.

Authors have well revised several issues. The following comments are addressed and require minor modifications to enhance the quality of the manuscript:

-I would begin with the title of the work which I kindly suggest being reviewed by the authors. The term “than”, in my opinion, does not seem well used.

-Please clarify in introduction the HbA1c level that defines type 2 DM diagnosis (line 40) and which periodontal treatments are included in supportive periodontal therapy (line 69)

-Clinical data collection:  I would like to know, if possible, about smoking history of participants.

-The calculation of PESA and PISA, by the report of Nesse et al, could better explained in methodology. (line 99)

- In conclusion mean PPD  (line 204) relates to higher/lower values of probing depths -please review this sentence.

Reviewer 2 Report

This is an interesting study. The authors may look into the following to improve the paper.

1.       What does SPT as defined in this study include. How did you ensure that all included patients were similar with respect to treatments received by them and number of appointments.

2.       Can the treatment being given for T2DM influence PISA.

3.       Provide details of recording PISA from PESA and BOP. Elaborate on these aspects.

4.       Was all this data available in the retrospective data from medical records.

5.       “…it is necessary to find a marker that can be used to determine the correlation between HbA1c levels and periodontitis.” This statement is not clear.

6.       What is the objective of the study and expected outcomes.

7.       Were other co-morbidities present in these patients apart from T2DM.

8.       Unable to understand 2.4 and 2.5: “Adjustments were made for the following variables: sex, age, number of teeth, and mean PPD in model 1; sex, age, number of teeth, and PISA in model 2.” 

As PISA depends on the PPD, why is it being correlated with it.

9.       Does PPD correlate better with HbA1C than PISA. Hence, is it a better clinical parameter than PISA. Are there any situations which warrant use of PISA?

10.   Does the study imply that residual pockets or deep pockets even during SPT are not associated with glycemic changes. Or Does it mean that improved pocket depths during SPT correlates with improvement of HbA1C levels. For how much of an increase in probing depth will there be a change in HbA1C level?

11.   It is not clear from the study whether HbA1C levels recorded were only during SPT or indicated changes in HbA1c from the period prior to initiation of SPT or from baseline.

12.   “…this retrospective cohort study clarified that participants who had developed glucose intolerance or diabetes during the past 10 years showed a significant increase in periodontal pocket depth on subsequent examination.”  Material and methods section can be elaborate to indicate at what time intervals these measurements were taken. How long was each patient followed up in the retrospective data.

13.   Line 191-192: “In contrast, PISA may predict HbA1c levels in patients with severe diabetes with extensive inflammation in the periodontal tissue [16,17].” This statement cannot be used to substantiate results of the present study, where there was no correlation between the two parameters even in severe periodontitis.

14.   Please comment on the association of other parameters examined with HbA1C also like gender etc. How were the oral hygiene practices of these patients. Were there abusive habits recorded during SPT. Was a prosthesis given during this period. Did the patients have more extractions during this period of review. Also, the number of years since diagnosed with diabetes, were they obese?

15.   Did you consider CAL or recession for measuring PESA. Does using PPD without considering CAL / recession lead to overestimation of PISA.

16.   BOP more accurately reflects the collagen, blood vessel density and epithelial thickness. Your methods of assessing PISA seem to be different from Nesse 2008. Please clarify.

17.   The criteria for inclusion into this study are not clear.

18.   Was a sample size estimation done. Was there normalized distribution of data.

19.   What was the PISA range for the mild and severe periodontitis groups.

Reviewer 3 Report

This paper aims to evaluate the relationship between clinical features of periodontal  disease with glycated hemoglobin in controlled periodontal patients. It is a well structured and organized document.

Title: the title is confusing. I suggest some modifications related to the objectives like “association between clinical periodontal features and glycated hemoglobin in diabetic patients with controlled periodontitis” or similar.

Abstract: This study does not evaluate correlation. It evaluates the relationship between variables in form of association. multiple linear regression analysis does not determine correlation but the association between variables. Replace the word, please.

Conclusions do not mention PISA scores nor their relation with glycated hemoglobin.

The phrase “The multiple regression analysis showed a significant association between mean PPD (β= 0.38, P=0.03) and HbA1c in patients with mild diabetes mellitus but not in severe cases.” and this phrase “HbA1c was associated with PPD among the patients with mild diabetes treated with SPT.” conclude the same in mild periodontal patients. Please review this.

Introduction: In line 47 the word “in contrast,…” confuses the reader that expects a similar relation between periodontal disease and glycated haemoglobin, which was explained in the previous sentence.

Substitute “correlation” for “association” between variables.

Objectives should be related to null hypothesis tested in the statistical analysis.

Material and Methods:

Explain how PISA and PESA were calculated since they are dependent variables on this study. Also explain better how to categorize the variables BOP and PBB and how they were obtained.

Statistical analysis should be perform after checking normality of the variables and results presented as medians or averages dependent on the normality test.

Explain the division of patients in mild and severe periodontitis.

Do not subtitle the statistical analysis chapter in subtitles, the text is repeated and is unnecessary to do so.

Results- table 1 has one item “mean PPD” that should be just PPD

Introduce results on how many patients were in each group (mild or severe periodontitis). Also, introduce BOP and PESA results.

Explain how the variables are associated…. When a number is high the other one is present… for example. Explain in the text what is relevant in the tables, not just introduce them.

Reviewer 4 Report

There is a mutual relationship between periodontitis and type 2 diabet. There are many important biomarkers used as a reference for clinical diagnostic criteria. Despite the authors' attempts to find new biomarkers between the severity of periodontitis and the level of glycated hemoglobin, they did not receive new data. In this regard, the authors advised themselves to take a group of patients with severe periodontitis and conduct further research. In my opinion, the article as presented is not a complete study and can only serve as an example of the use of multivariate linear regression  analysis in medical research. In addition, neither the introduction nor the discussion analyzed possible pathogenetic relationships between the studied parameters, the relationship with the microbiota and immune mechanisms.

Reviewer 5 Report

The article poses an interesting hypothesis however several issues must be addressed. Please see enclosed PDF

Round 2

Reviewer 2 Report

The authors may include the review response 18 on model 1 and 2 in the text.

Further, response no. 15 given by the authors on CAL may also be added into the text in the appropriate place. 

Reviewer 4 Report

The authors have done a good job and after making edits, the meaning of the article, the purpose and conclusions have become much clearer and more fully formulated. The article can be accepted for publication and hopes that the authors will continue working in this direction.
